# Implementation of a care manager organisation and its association with antidepressant medication patterns: a register-based study of primary care centres in Sweden

Christine Sandheimer ![ORCID],[1] Cecilia Björkelund ![ORCID],[1,2,3] Gunnel Hensing,[1] Kirsten Mehlig,[1,4] Tove Hedenrud[1]

For numbered affiliations see end of article.

**Correspondence to**
Ms Christine Sandheimer;
christine.sandheimer@gu.se

## ABSTRACT

**Objective** To evaluate the implementation of a care manager organisation for common mental disorders and its association with antidepressant medication patterns on primary care centre (PCC) level, compared with PCCs without this organisation. Moreover, to determine whether a care manager organisation is associated with antidepressant medication patterns that is more in accordance with treatment guidelines.

**Design** Register-based study on PCC level.

**Setting** Primary care in Region Västra Götaland, Sweden.

**Participants** All PCCs in the region. PCCs were analysed in three subgroups: PCCs with a care manager organisation during 2015 and 2016 (n=68), PCCs without the organisation (n=92) and PCCs that shifted to a care manager organisation during 2016 (n=42).

**Outcome measures** Proportion of inadequate medication users, defined as number of patients ≥18 years with a common mental disorder diagnosis receiving care at a PCC in the region during the study period and dispensed 1–179 defined daily doses (DDD) of antidepressants of total patients with at least 1 DDD. The outcome was analysed through generalised linear regression and a linear mixed-effects model.

**Results** Overall, all PCCs had about 30%–34% of inadequate medication users. PCCs with a care manager organisation had significantly lower proportion of inadequate medication users in 2016 compared with PCCs without (−6.4%, p=0.02). These differences were explained by higher proportions in privately run PCCs. PCCs that shifted to a care manager organisation had a significant decrease in inadequate medication users over time (p=0.01).

**Conclusions** Public PCCs had a more consistent antidepressant medication pattern compared with private PCCs that gained more by introducing a care manager organisation. It was possible to document a significant decrease in inadequate medication users, notwithstanding that PCCs in the region followed the guidelines to a comparatively high extent regardless of present care manager organisation.

## Strengths and limitations of this study

► This is the first study aimed at assessing the implementation of a care manager organisation in Swedish ordinary primary care (PC) on a regional level.

► Register data on dispensed antidepressant medication per PC centre (PCC) is a strong quality indicator in the assessment of the PCC's adherence to clinical guidelines.

► The use of register data made it possible to include all PCCs in the southwest of Sweden.

► As primary healthcare is context bound, the generalisability of the findings from this study may be limited.

► As the differences between the PCCs and over time were small, future studies should include a longer time period to be able to distinguish the long-term effects of a care manager organisation in Swedish PC.

## INTRODUCTION

Common mental disorders (CMDs) such as depression, anxiety syndromes, and adjustment and stress-related disorders are the major causes of the disease burden worldwide.[1] In high-income countries, such as most European countries and the USA, approximately 18% of the population experience a CMD during a year.[2] The economic burden of CMDs contain both direct costs, in the forms of costs of healthcare visits and medications, and indirect costs, such as sickness absence and production loss.[3]

When it comes to healthcare visits, individuals with CMDs often seek care at the primary care (PC) as the first contact with the healthcare system and it is in the PC the majority are treated.[4–6] In spite of this, for several years research has shown that the PC systematically

fails to identify patients with CMDs,[4] especially when the patient seek care for a somatic complaint.[7] Moreover, the treatments for CMDs, the most common being psychotherapy and antidepressant medication use, have been shown to be inadequate and to not always follow established clinical treatment guidelines.[8] A suboptimal use of antidepressant medication increases the risk of relapse and might adversely affect the treatment outcome.[9] Research has also shown that just starting patients on antidepressant medications without the support and continuity given in coordinated and collaborative care only helped a minority of patients with depression.[10]

During the last two decades, several countries have started to implement collaborative care models in their PC to improve the care given to patients with CMDs.[11 12] The collaborative care model is an organisational intervention often including four key components: (1) an interdisciplinary approach, (2) structured care planning, (3) a systematic follow-up of the patient and (4) increased communication between healthcare personnel.[13] The model was adapted from the Chronic Care Model of disease management developed by Wagner *et al*.[14] Wagner *et al* suggested that team-based care, incorporating the four above-mentioned components and where a non-physician caregiver (eg, a nurse) had the main responsibility for the patient, would make the care more efficient and would improve treatment outcomes.[14]

In 2012, the Swedish Council on Health Technology Assessment[15] stressed a need for studies in Swedish PC to determine whether the collaborative care model with special trained care managers could result in the same positive effects as shown in other PC settings.[16 17] Thus, a first pragmatic, cluster-randomised controlled trial (PRIM-CARE) with 23 PC centres (PCC) in Region Västra Götaland in the south-west of Sweden was conducted in 2014.[18] The results from the PRIM-CARE study showed positive effects of the care manager organisation regarding symptom remission and a more consistent antidepressant medication use compared with care as usual (CAU).[18] After the intervention study, a new phase was initiated with the aim of implementing a care manager organisation more widely in the region's PCCs.

Even though the positive effects on depression and anxiety outcomes of collaborative care in the PC have been well established by now, the evidence base regarding quality assessment of important components in the collaborative care model is poor.[19] One important process variable for a quality assessment is to measure the continuation of antidepressant medication use.[20] A common variable to measure continuity of medication is defined daily doses (DDD) defined as: 'The assumed averaged maintenance dose per day for a drug used for its main indication in adults.'[21] According to clinical guidelines, antidepressant medication use for depression and anxiety disorders should continue for at least 6 months (180 days) to minimise the risk of relapse of symptoms and unnecessary prolongation of the disorder.[22 23]

The present study is the first in Sweden that aimed to quality assess the care manager organisation in Swedish PC on PCC level by investigating the association between the implementation with antidepressant medication patterns. Our objectives were: (1) to investigate antidepressant medication patterns in PCCs with and without a care manager organisation in 2015 (before the implementation) and in 2016 (when the implementation was accomplished), (2) to investigate longitudinal changes in antidepressant medication patterns among the PCCs from 2015 to 2016 and (3) to determine whether establishing a care manager organisation is associated with antidepressant medication patterns more in accordance with established national treatment guidelines.

## METHODS
### Study design and context
The present study was based on aggregated register data at PCC level and was a collaboration between the research programme 'New Ways—mental health at work' and the research project 'PRIM-CARE - Care manager and collaborative care for common mental disorders in primary care' that is part of the research platform 'Ways-of-life, stress and mental health in the PC context', both at the School of Public Health and Community Medicine, University of Gothenburg, Sweden and Research and Development Primary Healthcare, Region Västra Götaland.

### Study setting and participating PCCs
This study was conducted in the south-western part of Sweden, in the Region Västra Götaland with a population of 1.7 million (17% of the Swedish population).

Region Västra Götaland is not just a geographical area in Sweden but also an organisational entity, responsible for financing and producing the healthcare in the region. For the costs of outpatient medications, the region receives an earmarked budget from the state. To ease the economic burden on the individual, the state introduced a cost-containment scheme with a rising scale of state subsidies (10%, 25%, 50% and 100%), and with a cost ceiling of kr2350 (approx. €220). The cost-ceiling protects the individual from further expenses of outpatient medications during a 12-month period.

The number of PCCs in the region has varied over the years but is around 200, with a small majority having public management. In this study, we included PCCs with an ongoing organisation during both 2015 and 2016 in order to have comparable groups in the statistical analyses.

### The care manager organisation
The care manager organisation involves the whole staff at the PCC. All personnel are given brief training on the treatment guidelines for CMDs and how to collaborate with the care manager nurse. The care manager organisation aims at improving the collaboration among personnel, and at providing enhanced patient

information and patient education and support about treatment and treatment options. The care manager is a designated nurse that has completed special training in care management for patients with CMDs, especially depression. The work process of the care manager is as follows: when the patient is diagnosed with CMD by the PC physician, a contact with the care manager nurse is established where a person-centred care plan is drawn up together with the patient. The care plan is followed up by regular telephone contacts. During the weekly contacts with the patient, that continue for around 3 months, the care manager assesses the patient's CMD symptoms, and supports and engages the patients in following their treatment, in taking their medications according to the prescription, supervising potential side effects of medications and deciding if there is a need to change the dosage or type of medicine. The care manager has close contact with the treating physician and involves other professionals, like psychotherapists, if needed.

The implementation process in Region Västra Götaland started in the fall of 2015, with the first nurses finishing their examination in January 2016. More details about the care manager function and organisation can be found in a previously published article by Björkelund *et al*[18].

### Care as usual

CAU for patients with CMDs, in Swedish PCCs, can consist of visits to different healthcare professionals (such as physicians, nurses, psychologists/psychotherapists or physiotherapists). The treatment should follow the guidelines set by the region which are based on the national evidence-based clinical guidelines for CMDs.[24] These guidelines include face-to-face psychotherapy, antidepressant medication treatment and/or sick listing.

### Data and study period

We collected register data on healthcare use and dispensed medications among patients aged 18 and older diagnosed with F32, F33, F40, F41 or F43, (ie, depression, anxiety syndrome, social phobia and/or stress induced mental ill health) who were receiving care at a PCC in the region within the study period. The data were collected on an aggregated level with information per PCCs and not per patient. The data were obtained from the healthcare database VEGA and the regional prescribed drug register Digitalis. Both are managed by Region Västra Götaland. VEGA contains healthcare information about the inhabitants in the region; all healthcare consumption is included even the care that has been consumed outside the region. Digitalis contains both information on dispensed medications prescribed in the region from one of the region's PCCs but dispensed in any Swedish pharmacy, as well as information about prescriptions dispensed by the region's inhabitants, even those that are prescribed outside the region.

The study period was January to December 2015 (before implementation of a care manager organisation) and January to December 2016 (when the implementation was accomplished).

### Outcome measure

The study population was based on number of patients aged 18 and older with a CMD diagnosis who were dispensed at least one antidepressant medication during the study period, as obtained from each PCC. Patients who were dispensed between 1 and 179 DDD of antidepressant medication (ie, less than 6 months treatment) were defined as inadequate medication users in accordance with clinical treatment guidelines for CMDs.[22 23] As outcome measure, we used the Proportion of inadequate medication users defined as number of patients with CMD who were dispensed between 1 and 179 DDD divided by the number of all CMD patients with were at least 1 dispensed DDD. Antidepressants (Anatomical Therapeutic Chemical code N06A) was the medication group of interest in this study as these medications are used in the treatment of both depression and anxiety syndromes.[25]

### Analytical sample and covariates

The included PCCs were divided into three groups based on their care manager status: (1) PCCs that finished the implementation process of a care manager organisation in January 2016 and continued having the organisation during all of 2016, (2) PCCs that shifted from not having a care manager organisation in 2015 to implementing the organisation from June 2016 and forward, and (3) PCCs without a care manager organisation during the study period (CAU). The following variables were included as potential confounders: (1) private or public management; (2) number of listed patients and (3) proportion of patients with a CMD diagnosis (number of patients with a CMD diagnosis divided by number of listed patients).

### Statistical analyses

The outcome variable proportion of inadequate medication users had a distribution that was skewed to the right, and a log-transformation of the variable yielded normally distributed residuals in linear regression models.

Group differences by care manager status were assessed with the non-parametric Kruskal-Wallis test and presented with p values.

Cross-sectional analyses used linear regression of log proportion of inadequate medication users on the categories of care manager status (ref=CAU), and further adjusted for private management (ref=public management). The resulting beta-coefficients β were further transformed to give the relative difference in percent for log proportion of inadequate medication users in PCCs with a care manager organisation compared with PCCs without, calculated as $(\exp(\beta) - 1) \times 100\%$. Cross-sectional analyses were performed for both 2015 and 2016.

To investigate the effect of care manager status on the longitudinal change in log proportion of inadequate medication users within PCC groups we implemented a

linear mixed-effects model. The model took into account the correlation between repeated measures for the same PCC.[26] The model for longitudinal change included care manager status, year, the interaction between care manager status and year, and further adjusted for private management status, number of listed patients and proportion of patients with CMD diagnosis.

Analyses were performed using IBM SPSS Statistics V,25 and 26, SAS (V.9.4; SAS Institute) and MATLAB (R2016b; The MathWorks). Statistical significance was set at 0.05 (two-sided tests) with presentation of 95% confidence intervals (CI).

### Patient and public involvement

Neither patients nor the public were involved in the design or conception of this study.

### RESULTS

Table 1 presents the description of PCCs by care manager status from baseline (2015) and after accomplished implementation (2016). A total of 190 PCCs were included in the study, of which 81 were classified as not having a care manager organisation (CAU), 42 as PCCs that shifted and 67 as having a care manager organisation. Public or private management (PCC status) was highly correlated with care manager status (p<0.001). The proportion of patients with a CMD diagnosis (of total number of listed patients) per PCC varied between 9% (2015) and 10% (2016). Visits to nurses and physicians by the study population were higher in 2016 compared with 2015 in all three PCC groups, but only visits to nurses had a statistically significant increase over time across the three categories of care manager status (p<0.01, data not shown). Overall, about 30%–34% of patients with a CMD diagnosis, in all three PCC groups, had less than 180 DDD of antidepressant medication. PCCs with a care manager organisation had the lowest proportion of inadequate medication users in both 2015 and 2016 compared with the other two PCC groups. All PCC groups had lower proportions of inadequate medication users in 2016 compared with 2015.

### Cross-sectional results

In both years, cross-sectional analyses showed that PCCs with a care manager organisation had about 6% lower proportion of inadequate medication users compared with PCCs without, but the difference was only significant in 2016 (p=0.02) (table 2).

After controlling for private or public management the differences by care manager status reduced. The covariates number of listed patients and proportion of patients with CMD diagnosis did not have any effect on the association between proportion of inadequate medication users and care manager status (data not shown). The association between care manager status and proportion of inadequate medication users did not differ between private or public PCCs (p value for interaction >0.4 at both time points).

### Longitudinal results

We used a linear mixed-effects model to investigate changes in log proportion of inadequate medication users between the three PCC groups with different care manager status over time (figure 1). The results showed no overall interaction between year and care manager status (p=0.17, data not shown). As illustrated in figure 1, all three PCC groups had a decrease in proportion of inadequate medication users over time but this change was only significant in PCCs that shifted from CAU to a care manager organisation (p=0.01). This trend was also observed after further adjustment for private management (p=0.01) and for the proportion of patients with a CMD diagnosis (p=0.02).

### DISCUSSION

The present study is the first in Sweden that, on PCC level, has assessed a new care function with an organisational change in the Swedish PC, that is, the care manager organisation.

The main finding of this study was that a significantly lower proportion of patients with inadequate antidepressant medication could be shown for the PCCs that had implemented a care manager organisation from the start compared with PCCs without. Furthermore, only PCCs that shifted from CAU in 2015 to a care manager organisation at some point during 2016 had a significant decrease of inadequate medication users over time. This PCC group was dominated by privately driven PCCs. Among PCCs with a care manager organisation from the start, the change was barely visible as these PCCs had a lower proportion of inadequate antidepressant medication users even before the implementation.

The absence of significant differences between the PCC groups could possibly be partially explained by the fact that PCCs in Region Västra Götaland are bound by contract to deliver care according to regional clinical guidelines. The region has set up prescribing objectives for a rational medication use.[27] These objectives are found in a prescribing guidelines booklet available for all PCCs to help them develop good prescribing routines. This could be interpreted to mean that even a PCC without a care manager organisation can deliver high pharmacological care quality that resonates well with the guidelines. Our assumption is supported by the findings from a study performed in the same region as our in which PC physicians' attitudes and use of the prescribing guidelines booklet were investigated.[28] The study found that the majority of physicians had positive attitudes towards the booklet and used it frequently. The same study also showed that physicians in private PCCs were more negative towards the booklet and did not adhere to the guidelines to the same extent as the physicians in the public organisations. This result could partly explain our finding that the public PCCs were stable in their medication use after the implementation of a care manager organisation and mostly the private PCCs showed an improvement.

**Table 1** Description of PCCs by care manager status (CAU n=81, Shift* n=42 or care manager organisation n=67) the year before implementation (2015) and the year when the implementation was accomplished (2016)

| | Before implementation (2015) | | | | Accomplished implementation (2016) | | | |
|---|---|---|---|---|---|---|---|---|
| | CAU | Shift* | CMO | | CAU | Shift | CMO | |
| | n (%) | n (%) | n (%) | P value | n (%) | n (%) | n (%) | P value |
| **PCC status** | | | | | | | | |
| Public | 37 (46) | 16 (38) | 51 (76) | <0.001 | 37 (46) | 16 (38) | 51 (76) | <0.001 |
| Private | 44 (54) | 26 (62) | 16 (24) | | 44 (54) | 26 (62) | 16 (24) | |
| **PCCs by number of listed patients** | | | | | | | | |
| <5000 | 18 (22) | 7 (17) | 13 (20) | | 18 (22) | 7 (17) | 12 (18) | |
| 5000 to <7000 | 14 (18) | 9 (21) | 7 (10) | | 12 (15) | 8 (19) | 7 (10) | |
| 7000 to <10.000 | 26 (32) | 17 (41) | 21 (31) | | 26 (32) | 18 (43) | 23 (35) | |
| ≥10.000 | 23 (28) | 9 (21) | 26 (39) | | 25 (31) | 9 (21) | 25 (37) | |
| Mean value (SD) | 8206 (3631) | 8268 (3786) | 8855 (3842) | 0.53 | 8452 (3594) | 8442 (3749) | 8921 (3802) | 0.69 |
| **PCCs by number of patients with CMD diagnosis** | | | | | | | | |
| <500 | 23 (28) | 10 (24) | 15 (22) | | 16 (19) | 10 (24) | 13 (20) | |
| 500 to <700 | 18 (23) | 13 (31) | 14 (21) | | 20 (25) | 6 (14) | 9 (13) | |
| 700 to <1000 | 17 (21) | 10 (24) | 20 (30) | | 20 (25) | 14 (33) | 20 (30) | |
| ≥1000 | 23 (28) | 9 (21) | 18 (27) | | 25 (31) | 12 (29) | 25 (37) | |
| Mean value (SD) | 756 (344) | 774 (411) | 818 (394) | 0.95 | 820 (366) | 858 (429) | 901 (429) | 0.25 |
| **PCCs by number of patients on antidepressants** | | | | | | | | |
| <300 | 23 (28) | 11 (26) | 13 (20) | | 17 (21) | 11 (26) | 11 (17) | |
| 300 to <500 | 24 (30) | 18 (43) | 23 (34) | | 30 (37) | 14 (34) | 21 (31) | |
| 500 to <700 | 21 (26) | 6 (14) | 20 (30) | | 18 (22) | 9 (21) | 18 (27) | |
| ≥700 | 13 (16) | 7 (17) | 11 (16) | | 16 (20) | 8 (19) | 17 (25) | |
| Mean value (SD) | 450 (223) | 461 (241) | 485 (233) | 0.70 | 487 (231) | 499 (249) | 528 (244) | 0.57 |
| **PCCs by number of visits by patients with CMD to:** | | | | | | | | |
| Physicians, n† | 80 | | 66 | | | | 67 | |
| <1000 | 19 (24) | 12 (29) | 14 (21) | | 17 (21) | 9 (21) | 9 (13) | |
| 1000 to <1500 | 22 (27) | 12 (29) | 15 (23) | | 14 (17) | 11 (26) | 15 (23) | |
| 1500 to <2000 | 15 (19) | 8 (19) | 13 (20) | | 17 (21) | 5 (12) | 11 (16) | |
| ≥2000 | 24 (30) | 10 (23) | 24 (36) | | 33 (41) | 17 (41) | 32 (48) | |
| Mean value (SD) | 1660 (855) | 1584 (994) | 1839 (954) | 0.27 | 1873 (946) | 1985 (1245) | 2202 (1218) | 0.35 |

Continued

**Table 1** Continued

| | Before implementation (2015) | | | | Accomplished implementation (2016) | | | |
| | CAU | Shift* | CMO | P value | CAU | Shift | CMO | P value |
| | n (%) | n (%) | n (%) | | n (%) | n (%) | n (%) | |
|---|---|---|---|---|---|---|---|---|
| Nurses, n† | 76 | 36 | 64 | | 77 | 37 | 67 | |
| <50 | 34 (45) | 15 (42) | 23 (36) | | 18 (23) | 9 (25) | 11 (17) | |
| 50 to <300 | 22 (29) | 15 (42) | 17 (26) | | 25 (33) | 19 (51) | 25 (37) | |
| 300 to <500 | 6 (8) | 3 (8) | 12 (19) | | 8 (10) | 2 (5) | 6 (9) | |
| ≥500 | 14 (18) | 3 (8) | 12 (19) | | 26 (34) | 7 (19) | 25 (37) | |
| Mean value (SD) | 293 (453) | 191 (322) | 291 (450) | 0.61 | 455 (588) | 282 (368) | 571 (690) | 0.05 |
| Proportion of inadequate medication users | | | | | | | | |
| % (SD) | 32.8 (9.7) | 33.8 (10.4) | 30.3 (5.1) | 0.15 | 32.5 (8.3) | 31.6 (5.3) | 29.9 (4.0) | 0.18 |

Text marked in bold is statistically significant at the 0.05% level (two-sided)
*Shifted from CAU to a care manager organisation.
†Number of valid observations.
CAU, care as usual; CMD, common mental disorder; CMO, care manager organisation; PCC, primary care centre.

Approximately 70% of the patients with a CMD diagnosis, in all three PCC groups, were dispensed an antidepressant medication for 6 months or longer (180 DDDs or more), which is in accordance with the recommendations in established clinical guidelines.[22 23] Findings from other studies have presented numbers between 55% (France) to 66% (Denmark), and 74% (Sweden) of antidepressant length lasting for more than 6 months after remission.[29–31] As an international comparison, Sweden has a high proportion of the population taking antidepressant medications,[32–34] and Region Västra Götaland stands out as one of the regions in Sweden with highest prevalence of antidepressant users.[35]

A point important to acknowledge is that not all individuals need a medication treatment that lasts more than 180 days. An ambition of 100% medication use in accordance with treatment guidelines is therefore unrealistic. Research has shown that about 15% of Swedish patients do not collect their medications and the reasons behind are many.[36] The guidelines should be seen as recommendations not disregarding the patients' individual needs.[36]

In the preceding intervention study (of the present study), the result showed that a care manager positively affected the antidepressant treatment in patients with depression.[18] The findings from the intervention study were in line with previous research from other healthcare contexts which has proved that support from a care manager during the treatment for CMDs increases the frequency of adequate antidepressant medication use on the individual level.[10 15 20 37]

## Methodological considerations

The chosen design for this study possesses both strengths and potential limitations. A strength with a register based study is the reduced risk of bias due to low sampling, high attrition levels, recall bias and other methodological fallacies that are more frequent in survey studies.[38] The register databases used in this study; the healthcare database VEGA and the regional prescription drug register Digitalis, are comprehensive, systematic and updated on a regular basis. Digitalis is based on the same data as the national prescribed drug register in Sweden, enabling a national comparison of data. The use of data from VEGA and Digitalis allowed us to include all PCCs in the region of various sizes and management types, and from different rural and urban settings. There are approximately 1200 PCCs in Sweden, and this makes our sample, and thus our findings, generalisable on a national level. However, the applicability of our results to an international context could be limited since healthcare systems, and particularly, the PC organisation, differ widely between countries.[15]

A limitation with the use of aggregated data is the risk of information bias that is difficult to control for when we lack detailed information on the individual level.[39] The registers contain comprehensive information but they also require skills and experience of researchers to discern what the statistics embody and if the information is managed correctly. In our case, a thorough scrutiny of

**Table 2** Association between log proportion of inadequate medication users and PCCs' care manager status (CAU, shift† or care manager organisation)

| | | 2015 (Baseline) | | 2016 | |
| | | Relative difference§ % (95% CI) | | Relative difference§ % (95% CI) | |
| | n‡ | Unadjusted model | Adjusted for private PCC | Unadjusted model | Adjusted for private PCC |
|---|---|---|---|---|---|
| CAU | 81 | Ref. | Ref. | Ref. | Ref. |
| Shift | 42 | 2.7% (–5% to 10.1%) | 1.6% (–5.6% to 9.4%) | –1.5% (–7.8% to 2.5%) | –2.5% (–8.3% to 3.7%) |
| CMO | 67 | –6.1% (–12.3% to 0.4%) | –2% (–8.3% to 4.7%) | –6.4% (–11.6% to –0.9%)* | –2.6% (–7.9% to 2.9%) |

*P<0.05.
†Shifted from CAU to a care manager organisation.
‡Number of valid observations.
§Relative difference in percent=(exp(β)−1)×100%.
CAU, care as usual; CMO, care manager organisation; PCC, primary care centre.

the data led to the exclusion of 22 PCCs that had extreme values of inadequate medication users because the PCCs did not have a healthcare activity (ie, were in operation) during the whole study period. As a result, we believe the reliability of our findings improved substantially.

Another methodological issue is the small observed differences between the PCC groups that could be due to the chosen period of study. The implementation process for the care manager organisation started in January 2016 but the education for the care manager nurses began in late 2015. To have used 2015 as the baseline year and 2016 as the follow-up year could perhaps have led to the absence of greater differences. A longer follow-up period of potential effects of the introduction of the care manager organisation would have been preferable.

The last potential limitation in the present study is the absence of prescription data, that is, information about what the physicians prescribed. A comparison between dispensed and prescribed medications to discern possible disparities in the data would have made a stronger quality indicator. Nevertheless, dispensed medication is a better indicator of medication use than prescription data[40] as it

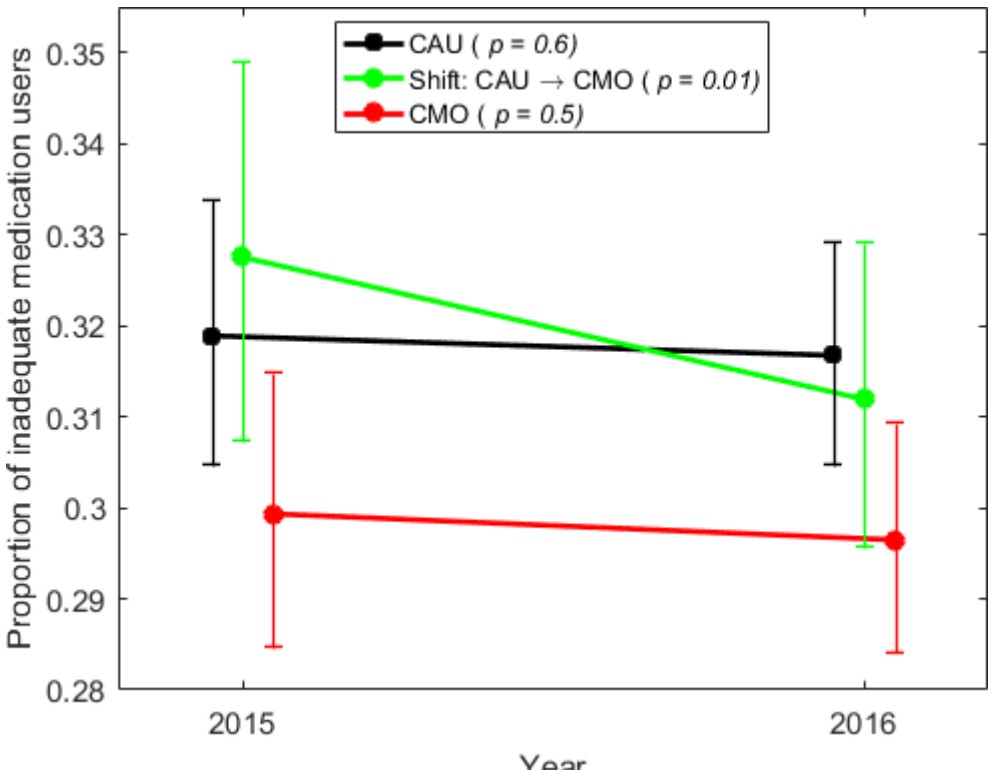

**Figure 1** Unadjusted model of geometrical mean values (95% CI) for proportion of inadequate medication users by category of care manager status (CAU, shift, CMO) and year. The p values indicate the significance of longitudinal changes within each category of care manager status. CAU, care as usual; CMO, care manager organisation.

gives us a clearer picture of the active behaviour of the patient rather than the prescriber, and because many patients do not fill their prescriptions.[36] Hence, we believe that register data on dispensed medication was the better option for measurement as one of the tasks of the care manager is to help the patients to take their medicines according to the prescription guidelines.

## Implications

Antidepressant medication treatment is influenced by several different factors that affect the outcome. These include both medical and patient-related factors such as accessibility, diagnostics and diagnostic process, guidelines, patient knowledge and adherence.[41 42] Most of these factors can be influenced by the care manager,[19] and thus, an assessment of quality improvements achieved by a care manager organisation, at the PCC, should be evaluated not only on the individual patient level, but also on the PCC level by using healthcare registers. Assessments of complex interventions aiming at changing routine practice in PC form an important knowledge base for decision-makers responsible for the provision and financing of healthcare.

As this is the first study that assessed the implementation of a care manager organisation in Swedish PC on PCC level with data when the organisational change was new, future studies are needed to further investigate the effects of the care manager organisation on a long-term basis.

## CONCLUSIONS

This quality assessment of the care manager organisation for patients with CMDs in Swedish PC constitutes an important knowledge base in the further implementation process. The results of this study showed that public PCCs had consistently lower numbers of inadequate medication users compared with PCCs with private management, that gained more in introducing a care manager organisation. Moreover, we showed that PCCs in Region Västra Götaland, in general, followed the guidelines to a comparatively high extent both with and without the introduction of a care manager organisation. However, future studies should include a longer time period to be able to distinguish the long-term effects of a care manager organisation in Swedish PC.

**Author affiliations**
$^1$Social medicine, School of Public Health and Community Medicine, Institute of Medicine, The Sahlgenska Academy, University of Gothenburg, Goteborg, Sweden
$^2$Primary Health Care, School of Public Health and Community Medicine, Institute of Medicine, The Sahlgenska Academy, University of Gothenburg, Gothenburg, Sweden
$^3$Research and Development, Primary Health Care, Region Västra Götaland, Sweden
$^4$Lifecourse Epidemiology, School of Public Health and Community Medicine, Institute of Medicine, The Sahlgenska Academy, University of Gothenburg, Gothenburg, Sweden

**Acknowledgements** The authors would like to thank Nashmil Arai for her excellent work on the data management.

**Contributors** CB is the principal investigator and the initiator and guarantor of the project. CB, TH and CS participated in the design of the study. CS, CB, GH, KM and TH took part in planning the analyses. CS handled the data and had the main responsibility for conducting the analyses and for the writing of the paper. KM supervised and assisted in the statistical analysis. CB, GH, KM and TH contributed with the draft and revision of the manuscript. All authors contributed to the interpretation of data and read and approved the final version of this manuscript.

**Funding** This work was supported by the Swedish Research Council for Health, Working Life and Welfare (FORTE), grant number 2013-2216, and by grants from the Swedish state under the agreement between the Swedish government and the regions, the ALF-agreement (ALFGBG-722441) and Region Västra Götaland.

**Disclaimer** The funding organ had no influence on the design of the study, data collection, analysis or interpretation of data or writing of the report or the decision to publish the article. All authors were independent from the funder.

**Competing interests** None declared.

**Patient and public involvement** Patients and/or the public were not involved in the design, or conduct, or reporting, or dissemination plans of this research.

**Patient consent for publication** Not required.

**Ethics approval** The collected register data used in this study was on an aggregated level with information per PCC and not per patient. No individuals could be identified and no informed consent was necessary. Ethical approval was received from the regional Ethical Review Board in Gothenburg, Sweden (Dnr: T566-17).

**Provenance and peer review** Not commissioned; externally peer reviewed.

**Data availability statement** Data are available on reasonable request. Data are not publicly available due to Swedish law but are available from the authors on reasonable request.

**ORCID iDs**
Christine Sandheimer http://orcid.org/0000-0002-2429-0669
Cecilia Björkelund http://orcid.org/0000-0003-4083-7342

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
