## [Reviewer comments · BMJ Open]

ARTICLE DETAILS

TITLE (PROVISIONAL)	Implementation of a care manager organisation and its association with antidepressant medication patterns – a register-based study of primary care centres in Sweden
AUTHORS	Sandheimer, Christine; Björkelund, Cecilia; Hensing, Gunnel; Mehlig, Kirsten; Hedenrud, Tove

VERSION 1 – REVIEW

REVIEWER	Kaj Sparle Christensen Aarhus University, Denmark
REVIEW RETURNED	24-Nov-2020

GENERAL COMMENTS	1) P6L110-P7L18: please provide readers with a short and clear description of study objectives (in alignment with study design, analyses, findings and conclusions). 2) P2L21-L25: Please rephrase objective to be aligned with outcomes, results and conclusions. 3) What predefined hypotheses were actually tested in this study? Did authors expect that CMO would affect the numbers of patients with CMD diagnoses, the number of patients receiving antidepressants, or the number of visits to physicians or nurses? Was PIMU considered a primary or secondary analysis of data? P10L199-P11217: I suggest that you describe the statistical analyses performed without referring to results presented in tables and figures. Results should only be presented and referred to in the results section. P14L265-273: Summary of findings must be aligned with objectives and outcomes as mentioned previously. Findings displayed in table 1 show sparse evidence of CMO effects. Do you consider the statistically significant effect of CMO on PIMU to be clinically important? P18L361-368: again, please align conclusion with objectives and findings. Please fill in the Strobe checklist as supplementary material.
---

REVIEWER	Carlotta Lunghi Université du Québec à Rimouski, Canada
REVIEW RETURNED	30-Nov-2020

GENERAL COMMENTS	I thank the authors and the editor for the opportunity to comment on
--

the manuscript “Antidepressant medication use after implementation of a care management organization – a register-based study of primary care centres in Sweden”. I think the paper is interesting and addresses an important topic on antidepressant drug utilization research. I have only a few comments.

In general, I think there are too many abbreviations, which sometimes make the paper hard to read. I suggest the authors use only some of them in the text (i.e., primary care and primary care centers).

Lines 141-143: It is not clear to me if the collaborative care is implemented at the PCC level or the patient level, as suggested by these lines.

Lines 164-170: Definition of the outcome.

- I think the outcome is not well described by the definition proportion of inadequate medication use (PIMU). The measure is the proportion of inadequate medications USERS in the study year since we are at the PCC level and not the patient level. The authors did not measure “antidepressant adherence” through DDDs, but the proportion of users that received at least 180 DDDs. The definition should be changed through the paper.

- Line 166: I suggest the authors replace dispensed with claimed or received and reformulate the phrase to better understand.

- I am also concerned by the definition of inappropriate use as being exposed to 0 to 179 DDDs. Why did the authors choose to include 0? If a patient did not receive any antidepressants, why this should define as inappropriate? The patient may have been in psychotherapy instead of pharmacotherapy. The authors stated that they not have prescriptions data, so they cannot know if the patients did not claim any antidepressant because he/she did not receive a prescription or if he/she did not buy it at the pharmacy. I thus think that PIMU calculation should only include those patients who claimed at least one prescription (with a minimum DDD depending on the duration of the shortest prescription).

Lines 282-284: I think the sentence is not clear. What 70% refer to? Do the authors mean that approximately 70% of patients in the three PCCs groups had a medication used following guidelines (notably 180 DDDs and above)? This means that the majority of patients used antidepressants for at least 6 months. Or 70% of PCCs had a high proportion (which proportion?) of “good users” (180 DDDs or more)? If the good interpretation is the first one, I think the result is peculiar, since, to my knowledge, a large part of antidepressant users stops their medications early before 6 months. For instance, in a recent study my colleagues and I did in Italy (Lunghi C, Antonazzo IC, Burato S, et al. Prevalence and Determinants of Long-Term Utilization of Antidepressant Drugs: A Retrospective Cohort Study. *Neuropsychiatr Dis Treat.* 2020;16:1157-1170), a little more than 50% used an antidepressant only in the first year and about 45% claimed less than 180DDDs (results not shown in this article). I think the authors should add in the discussion a paragraph regarding this point. Are these results similar to what is known adherence to antidepressants is in Sweden?

VERSION 1 – AUTHOR RESPONSE

Reviewer 1: Dr Kaj Sparle Christensen

1) P6L110-P7L18: please provide readers with a short and clear description of study objectives (in alignment with study design, analyses, findings and conclusions).

RESPONSE: The paragraph has been restructured and rewritten.

2) P2L21-L25: Please rephrase objective to be aligned with outcomes, results and conclusions.

RESPONSE: The paragraph has been restructured and rewritten.

3) What predefined hypotheses were actually tested in this study? Did authors expect that CMO would affect the numbers of patients with CMD diagnoses, the number of patients receiving antidepressants, or the number of visits to physicians or nurses? Was PIMU considered a primary or secondary analysis of data?

RESPONSE: As this was an ecological study on aggregated level no predefined hypotheses were tested, the aim was not to falsify a null-hypothesis rather to observe and describe associations between the care manager organisation with the outcome measures. Our primary analysis of data was *Proportion of inadequate medication users (PIMU)*. Number of visits to physicians and nurses, as well as number of patients with CMD diagnosis and number of listed patients per PCC, were only considered as descriptive variables.

4) P10L199-P11217: I suggest that you describe the statistical analyses performed without referring to results presented in tables and figures. Results should only be presented and referred to in the results section.

RESPONSE: The paragraph has now been changed in accordance with the suggestions.

5) P14L265-273: Summary of findings must be aligned with objectives and outcomes as mentioned previously. Findings displayed in table 1 show sparse evidence of CMO effects. Do you consider the statistically significant effect of CMO on PIMU to be clinically important?

RESPONSE: The paragraph has been restructured and rewritten. We do believe our findings to be clinically important since they show a better adaptation to evidence and medical recommendations. The Swedish National Board of Health and Welfare (Socialstyrelsen) have for more than a decade stated that the antidepressant medication treatment should continue for at least 6 months. In Sweden, a gradual transference of responsibility for patients with CMD have taken place from the specialist psychiatry to the PC. However, the application of recommended treatment according to guidelines have been slow in the PC. Moreover, since the numbers are on an aggregated level, even small differences represent a substantial number of patients.

6) P18L361-368: again, please align conclusion with objectives and findings.

RESPONSE: The paragraph has been restructured and rewritten.

Reviewer 2: Dr Carlotta Lunghi

In general, I think there are too many abbreviations, which sometimes make the paper hard to read. I suggest the authors use only some of them in the text (i.e., primary care and primary care centers).

RESPONSE: We have tried to improve the readability by excluding the majority of abbreviations, however, established abbreviations such as PC for primary care, PCC for primary care centres, CAU for care as usual, and CMDs for common mental disorders we have chosen to keep.

Lines 141-143: It is not clear to me if the collaborative care is implemented at the PCC level or the patient level, as suggested by these lines.

RESPONSE: Collaborative care is implemented on PCC level. The paragraph has been restructured and changes have been made for clarification.

Lines 164-170: Definition of the outcome.

- I think the outcome is not well described by the definition proportion of inadequate medication use (PIMU). The measure is the proportion of inadequate medications USERS in the study year since we are at the PCC level and not the patient level. The authors did not measure “antidepressant adherence” through DDDs, but the proportion of users that received at least 180 DDDs. The definition should be changed through the paper.

RESPONSE: We are sincerely thankful for this suggestion. It was a bit of a hardship to not fall back in the patient level when trying to explain the outcome. The application of USERS instead of USE would be more correct.

- Line 166: I suggest the authors replace dispensed with claimed or received and reformulate the phrase to better understand.

RESPONSE: The phrase has been reformulated for clarification, however, claimed or received does not fit well with how the Swedish pharmaceutical system works. For a better understanding, a describing text about the system has been added (page 6-7 under study setting) and some linguistic adjustments have been made to the text.

- I am also concerned by the definition of inappropriate use as being exposed to 0 to 179 DDDs. Why did the authors choose to include 0? If a patient did not receive any antidepressants, why this should define as inappropriate? The patient may have been in psychotherapy instead of pharmacotherapy. The authors stated that they not have prescriptions data, so they cannot know if the patients did not claim any antidepressant because he/she did not receive a prescription or if he/she did not buy it at the pharmacy. I thus think that PIMU calculation should only include those patients who claimed at least one prescription (with a minimum DDD depending on the duration of the shortest prescription).

RESPONSE: Again, we are sincerely grateful for this scrutiny. Looking back at the raw data and the data extraction order, the data concerns patients that were dispensed at least one antidepressant during the study period. An error in the naming of the variable slipped in during the data management. Thus, 0-179 DDD is wrong, it should be 1-179, this has now been corrected.

• Lines 282-284: I think the sentence is not clear. What 70% refer to? Do the authors mean that approximately 70% of patients in the three PCCs groups had a medication used following guidelines (notably 180 DDDs and above)? This means that the majority of patients used antidepressants for at least 6 months. Or 70% of PCCs had a high proportion (which proportion?) of “good users” (180 DDDs or more)? If the good interpretation is the first one, I think the result is peculiar, since, to my knowledge, a large part of antidepressant users stops their medications early before 6 months. For instance, in a recent study my colleagues and I did in Italy (Lunghi C, Antonazzo IC, Burato S, et al. Prevalence and Determinants of Long-Term Utilization of Antidepressant Drugs: A Retrospective Cohort Study. *Neuropsychiatr Dis Treat.* 2020;16:1157-1170), a little more than 50% used an antidepressant only in the first year and about 45% claimed less than 180DDDs (results not shown in this article). I think the authors should add in the discussion a paragraph regarding this point. Are these results similar to what is known adherence to antidepressants is in Sweden?

RESPONSE: We meant that approximately 70 % of the patients with CMD that had dispensed an antidepressant during the study period, did so for at least 180 days. We thank you for your feedback. The sentence has been changed for clarification and a new paragraph has been added with additional references.

VERSION 2 – REVIEW

REVIEWER	Kaj Sparle Christensen Aarhus University, Denmark
REVIEW RETURNED	16-Feb-2021

GENERAL COMMENTS	Authors have adapted the manuscript in accordance with my instructions.
---

REVIEWER	Carlotta Lunghi Université du Québec à Rimouski
REVIEW RETURNED	18-Feb-2021

GENERAL COMMENTS	The authors have responded to all my concerns, and I think the paper has improved during the review.
--